# Constrained Diffusion with Trust Sampling

**William Huang**
Stanford University
willsh@stanford.edu

**Yifeng Jiang**
Stanford University
yifengj@stanford.edu

**Tom Van Wouwe**
Stanford University
tvwouwe@stanford.edu

**C. Karen Liu**
Stanford University
karenliu@cs.stanford.edu

## Abstract

Diffusion models have demonstrated significant promise in various generative tasks; however, they often struggle to satisfy challenging constraints. Our approach addresses this limitation by rethinking training-free loss-guided diffusion from an optimization perspective. We formulate a series of constrained optimizations throughout the inference process of a diffusion model. In each optimization, we allow the sample to take multiple steps along the gradient of the proxy constraint function until we can no longer trust the proxy, according to the variance at each diffusion level. Additionally, we estimate the state manifold of diffusion model to allow for early termination when the sample starts to wander away from the state manifold at each diffusion step. Trust sampling effectively balances between following the unconditional diffusion model and adhering to the loss guidance, enabling more flexible and accurate constrained generation. We demonstrate the efficacy of our method through extensive experiments on complex tasks, and in drastically different domains of images and 3D motion generation, showing significant improvements over existing methods in terms of generation quality. Our implementation is available at `https://github.com/will-s-h/trust-sampling`.

## 1 Introduction

Diffusion models are a class of generative models that have been highly successful at modeling complex domains, ranging from the generations of images [22, 13] and videos [24], to 3D geometries [32, 54, 4] and 3D human motion [52, 50], outperforming other deep generative models, such as GANs and VAEs [50, 13, 23]. Originally for unconditional generation, Diffusion models soon became used for cross-domain conditioned generation, such as text-conditioned image generations [43, 39], and generating human movements from audio [4].

For more fine-grained conditional generation where the samples need to precisely follow specified constraints, such as generating images following a certain contour, high-level controls like text prompts become insufficient. Guided diffusion has recently emerged to be a powerful paradigm on a variety of such constraints. One category of guided diffusion uses a separately trained classifier (as in classifier guidance [13]) or the score of a conditional diffusion model in classifier-free guidance [21]. For new constraints, the classifier or the conditional diffusion model must be retrained [60, 42, 57].

Alternatively, one can use the gradient of a loss function representing a constraint as guidance to achieve conditional diffusion [47, 11]. This flexible paradigm allows various constraints to be applied on a pre-trained diffusion model without compute cost on extra training. On this front, since the seminal works of Chung et al. [11] and Ho et al. [24], a number of techniques have been proposed to improve the quality of loss-guided diffusion, such as better step size design [58], multi-point MCMC

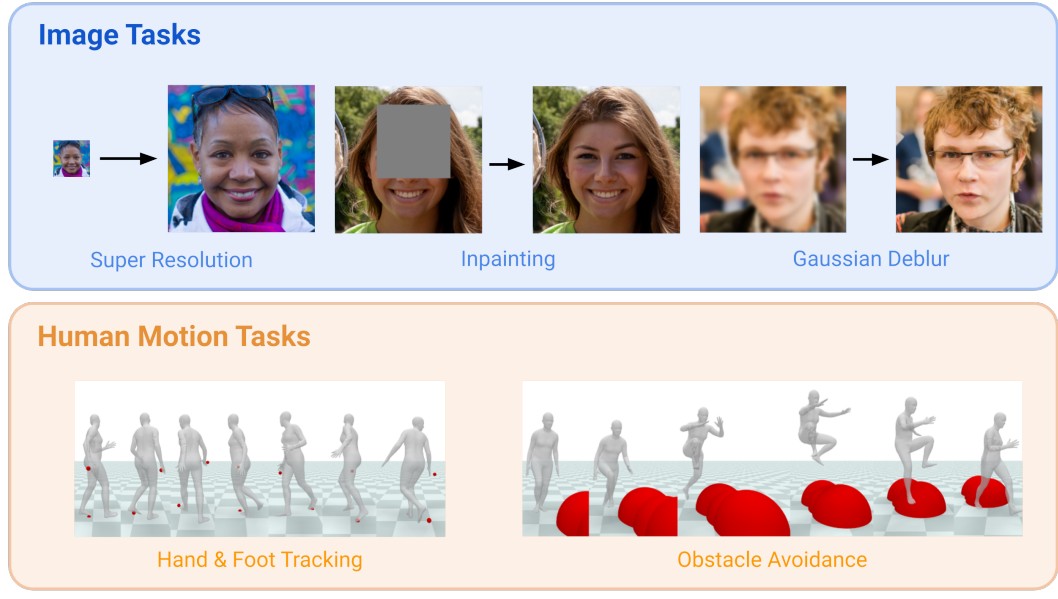

Figure 1: Trust Sampling can be applied to complex constraint problems in drastically different domains.

approximation [47], and incorporation of measurement models [46]. Several challenges remain for the current paradigm when trying to apply loss-guided diffusion for challenging constraints. For one, performance drops significantly when using a smaller budget of inference computation with fewer neural function evaluations (NFEs) [11, 58]. The methods are also sensitive to initialization, where previous evaluations often times take the best of a few generated samples for each constraint input.

In light of these challenges in training-free guided diffusion, we introduce Trust Sampling, a novel method that strays from the traditional approach of alternating between diffusion steps and loss-guided gradient steps in favor of a more general approach, considering each timestep as an independent optimization problem. Trust Sampling allows for multiple gradient steps on a proxy constraint function at each diffusion step, while scheduling the termination of the optimization when the proxy cannot be trusted anymore. Additionally, Trust Sampling estimates the state manifold of the diffusion model to allow for early termination, if the predicted noise magnitude of the sample exceeds the expected one in each diffusion step. Our framework is flexible, efficient, and performs well, achieving higher quality across widely different domains (e.g. human motion and images). We demonstrate the generality of Trust Sampling across a large number of image tasks (super-resolution, box inpainting, Gaussian deblurring) and motion tasks (root trajectory tracking, hand-foot trajectory tracking, obstacle avoidance, etc.). When compared to existing methods, we find that Trust Sampling satisfies constraints better and achieves higher fidelity.

## 2   Background

**Diffusion models.**    There are several equivalent formulations for diffusion models used in literature. Here, we briefly offer background on the denoising diffusion probabilistic model (DDPM) [22] formulation. Beginning from the data distribution $\mathbf{x}_0 \sim p(\mathbf{x})$, we can use a variance schedule $\beta_1, \ldots, \beta_T$ to produce latent variables $\mathbf{x}_1, \ldots, \mathbf{x}_T$ through the forward diffusion process $q(\mathbf{x}_t|\mathbf{x}_0) = \mathcal{N}(\sqrt{\alpha_t}\mathbf{x}_0, (1-\alpha_t)\mathbf{I})$, where $\alpha_t := \prod_{s=1}^{t}(1-\beta_s)$. In turn, a de-noising model $\boldsymbol{\epsilon}_\theta$ can be trained by minimizing the following loss function, which is a re-weighting of the variational lower bound [22]:

$$\mathcal{L}(\theta) = \mathbb{E}_{t,\mathbf{x}_0,\boldsymbol{\epsilon}}\left[||\boldsymbol{\epsilon} - \boldsymbol{\epsilon}_\theta(\mathbf{x}_t, t)||^2\right], \tag{1}$$

where $\mathbf{x}_0 \sim p(\mathbf{x})$, $t \sim \text{Unif}\{1, \ldots, T\}$, $\boldsymbol{\epsilon} \sim \mathcal{N}(\mathbf{0}, \mathbf{I})$, and $\mathbf{x}_t = \sqrt{\alpha_t}\mathbf{x}_0 + \sqrt{1-\alpha_t}\boldsymbol{\epsilon}$. The diffusion model can then be sampled in the reverse process, via DDPM [22] or the denoising diffusion implicit model (DDIM) formulation [45]:

$$\mathbf{x}_{t-1} = \sqrt{\alpha_{t-1}}\hat{\mathbf{x}}_0(\mathbf{x}_t) + \sqrt{1 - \alpha_{t-1} - \sigma_t^2}\boldsymbol{\epsilon}_\theta(\mathbf{x}_t, t) + \sigma_t\mathbf{z}, \tag{2}$$

where $\mathbf{z} \sim \mathcal{N}(\mathbf{0}, \mathbf{I})$ in both DDIM and DDPM, whereas $\sigma_t = \sqrt{(1 - \alpha_{t-1})/(1 - \alpha_t)}\sqrt{1 - \alpha_t/\alpha_{t-1}}$ is fixed in DDPM and can be chosen freely in DDIM. $\hat{\mathbf{x}}_0(\mathbf{x}_t)$ denotes the predicted $\mathbf{x}_0$ at timestep $t$, and can be written as

$$\hat{\mathbf{x}}_0(\mathbf{x}_t) = \frac{1}{\sqrt{\alpha_t}}(\mathbf{x}_t - \sqrt{1 - \alpha_t}\boldsymbol{\epsilon}_\theta(\mathbf{x}_t, t)). \tag{3}$$

Notably, DDPM and DDIM sampling can also be thought of a special case of gradient-based MCMC sampling (or a probability flow, in cases of DDIM without noise), where the goal is to refine the starting sample at each level $\mathbf{x}_t$ towards maximizing the likelihood $\mathbf{x}_{t-1} \sim p(\mathbf{x}_{t-1})$. In the case of DDPM/DDIM, instead of taking multiple MCMC steps [47] following the score function, only one step is taken at each level.

**Training-free Guided Diffusion.** One important application of Diffusion models is controlled (guided) generation. Instead of sampling from the unconditional data distribution $p(\mathbf{x})$, the goal is to sample from $p(\mathbf{x}|\mathbf{y})$, where $\mathbf{y}$ is the usually under-specified guidance signal. For example, an animator may wish to use an unconditional diffusion model of human motion $p(\mathbf{x})$ to generate motions with a constraint $\mathbf{y}$ that the character's right hand reaches to a specific location. Previous works [13, 11] transform the maximization of $p(\mathbf{x}_t|\mathbf{y})$ at each diffusion level $t$ with Bayes' rule:

$$\nabla_{\mathbf{x}_t} \log p(\mathbf{x}_t|\mathbf{y}) = \nabla_{\mathbf{x}_t} \log p(\mathbf{x}_t) + \nabla_{\mathbf{x}_t} \log p(\mathbf{y}|\mathbf{x}_t), \tag{4}$$

where we note that $\nabla_{\mathbf{x}_t} \log p(\mathbf{y}) = 0$. Existing algorithms therefore alternate between following the score function of the trained Diffusion $\nabla_{\mathbf{x}_t} \log p(\mathbf{x}_t)$, and following the guidance gradient $\nabla_{\mathbf{x}_t} \log p(\mathbf{y}|\mathbf{x}_t)$. However, directly optimizing $p(\mathbf{y}|\mathbf{x}_t)$ is generally intractable [11], as can be seen by the following probability factorization:

$$p(\mathbf{y}|\mathbf{x}_t) = \int_{\mathbf{x}_0} p(\mathbf{x}_0|\mathbf{x}_t)p(\mathbf{y}|\mathbf{x}_0)d\mathbf{x}_0, \tag{5}$$

where we used the fact that given $\mathbf{x}_0$, $\mathbf{y}$ is conditionally independent from $\mathbf{x}_t$. In general approximating $p(\mathbf{x}_0|\mathbf{x}_t)$ requires many denoising iterations of the Diffusion model, which is impractical when needing to alternate with optimizing $p(\mathbf{x}_t)$.

To address this difficulty, previous works [11, 58] approximate $p(\mathbf{y}|\mathbf{x}_t)$ with $p(\mathbf{y}|\hat{\mathbf{x}}_0(\mathbf{x}_t))$. Their observation is that in many practical applications, practitioners do have access to a closed-form differentiable function $L(\mathbf{x}_0, \mathbf{y})$ that can measure how good a clean (predicted or ground-truth) sample matches the desired condition $\mathbf{y}$. For example, $L$ can simply be the mean-squared error between target and actual positions of the right hand in our aforementioned animation application. Technically, by defining $L(\mathbf{x}_0, \mathbf{y})$ such that $p(\mathbf{y}|\mathbf{x}_0) \propto \exp(-L)$, $p(\mathbf{y}|\hat{\mathbf{x}}_0(\mathbf{x}_t))$ can be maximized by following the gradient direction $-\nabla_{\mathbf{x}_t} L(\hat{\mathbf{x}}_0, \mathbf{y})$.

Such frameworks open the door for highly flexible guided diffusion. Using the same unconditional model trained for $p(\mathbf{x})$, we can now plug in various different $L$ for different $\mathbf{y}$ during inference time, without having to train additional networks for each possible new $\mathbf{y}$.

## 3 Trust Sampling: Formulating Guided Diffusion as Optimization

Our work revisits training-free guided Diffusion from the perspective of optimization. Previous works decouple the two terms $p(\mathbf{x}_t)$ and $p(\mathbf{y}|\mathbf{x}_t)$ in Eq. 4 - they use the unconditional Diffusion model to optimize for $p(\mathbf{x}_t)$, and then took one gradient step of $\log p(\mathbf{y}|\mathbf{x}_t)$ for the constraint (guidance) term. As our experiment results will demonstrate, single gradient steps for constraints can lead to less optimal samples. With previous works mitigating this issue by carefully selecting the step sizes of $\nabla_{\mathbf{x}_t} \log p(\mathbf{y}|\mathbf{x}_t)$ [58] or by better approximating $p(\mathbf{y}|\mathbf{x}_t)$ [47], we explore a new direction which leads to a robust practical algorithm across multiple domains and various constraint diffusion tasks. To start with, following the gradients of $\log p(\mathbf{y}|\mathbf{x}_t)$ indicates we can reformulate the constraint diffusion problem as an optimization problem:

$$\max_{\mathbf{x}'} p(\mathbf{y}|\mathbf{x}') \quad \text{subject to } \mathbf{x}' \sim p(\mathbf{x}_t), \tag{6}$$

where we replace $\mathbf{x}_t$ with $\mathbf{x}'$ to signify that the state variable $\mathbf{x}'$ can deviate from the Diffusion-predicted $\mathbf{x}_t$ during this optimization. It is important to note that, first, for optimizing $p(\mathbf{x}_t)$, we still follow standard diffusion inference given its widespread empirical success in multiple domains, and

second, we constrain $\mathbf{x}'$ to stay in the distribution of all possible $\mathbf{x}_t$ at diffusion level $t$, as to not create a train-test discrepancy for the base diffusion model. This optimization formulation opens the door for more more flexibility in algorithm design, as we are no longer limited to taking only one gradient step.

## 3.1 Trust Schedules: Termination Criteria of Optimization

Our key improvement of this work is the use of iterative gradient-based optimization to solve Eq. 6. While only taking one single gradient step proves to be sub-optimal, as we will demonstrate in this section, optimizing until the objective saturates is also not ideal. To see this, recall that $p(\mathbf{y}|\mathbf{x}_t)$ is generally not tractable, while $p(\mathbf{y}|\hat{\mathbf{x}}_0(\mathbf{x}_t))$ is. As we replace the optimization objective $p(\mathbf{y}|\mathbf{x}_t)$ with the proxy $p(\mathbf{y}|\hat{\mathbf{x}}_0(\mathbf{x}_t))$, it is crucial to terminate in time before the proxy becomes a poor approximation of the true objective. Formally this relaxation can be written as:

$$\min_{\mathbf{x}'} \ L(\hat{\mathbf{x}}_0(\mathbf{x}'), \mathbf{y}) \quad \text{subject to } \mathbf{x}' \sim p(\mathbf{x}_t), \quad |p(\mathbf{y}|\mathbf{x}') - p(\mathbf{y}|\hat{\mathbf{x}}_0)| < d, \tag{7}$$

where readers are reminded that minimizing $L(\hat{\mathbf{x}}_0, \mathbf{y})$ is equivalent to maximizing $p(\mathbf{y}|\hat{\mathbf{x}}_0)$, and $d$ is a relaxation threshold newly introduced. To reason about the gap $d$ between true and proxy objectives, note that:

$$p(\mathbf{y}|\mathbf{x}') \quad = \quad \int_{\mathbf{x}_0} p(\mathbf{x}_0|\mathbf{x}')p(\mathbf{y}|\mathbf{x}_0)d\mathbf{x}_0 = \mathbb{E}_{\mathbf{x}_0 \sim p(\mathbf{x}_0|\mathbf{x}')}[f(\mathbf{x}_0)],$$

$$p(\mathbf{y}|\hat{\mathbf{x}}_0) \quad = \quad f(\hat{\mathbf{x}}_0) = f(\mathbb{E}_{\mathbf{x}_0 \sim p(\mathbf{x}_0|\mathbf{x}')}[\mathbf{x}_0]),$$

where we use $f(\cdot)$ as a shorthand for $\exp(-L(\cdot\,;\mathbf{y}))$. While a similar but tedious analysis exist for general multivariant $\mathbf{x}$, for the purpose of practical algorithm design, looking at the special case where $\mathbf{x}$ is a scalar random variable is more intuitive for understanding. Let $f''$ denote the curvature of $f(\mathbf{x})$, and $a = \inf f''$ and $b = \sup f''$ denote the range of the curvature assuming $\mathbf{x}$ has a finite span, we now have $f - \frac{1}{2}a\mathbf{x}^2$ and $\frac{1}{2}b\mathbf{x}^2 - f$ both as convex functions. Applying Jensen's inequality to both functions:

$$\mathbb{E}\left[f(\mathbf{x}_0) - \frac{1}{2}a\mathbf{x}_0^2\right] \geq f(\mathbb{E}[\mathbf{x}_0]) - \frac{1}{2}a\mathbb{E}[\mathbf{x}_0]^2, \quad \mathbb{E}\left[\frac{1}{2}b\mathbf{x}_0^2 - f(\mathbf{x}_0)\right] \geq \frac{1}{2}b\mathbb{E}[\mathbf{x}_0]^2 - f(\mathbb{E}[\mathbf{x}_0]). \tag{8}$$

After rearranging this gives:

$$\frac{a}{2}\text{Var}(\mathbf{x}) \leq \mathbb{E}[f(\mathbf{x}_0)] - f(\mathbb{E}[\mathbf{x}_0]) \leq \frac{b}{2}\text{Var}(\mathbf{x}). \tag{9}$$

This indicates, rather intuitively, that the approximation error increases with the variance of $\mathbf{x}$. As a result, we can **trust** the proxy optimization $\min_{\mathbf{x}'} \ L(\hat{\mathbf{x}}_0(\mathbf{x}'), \mathbf{y})$ more when the variance of $\mathbf{x}$ is smaller and the proxy becomes less reliable when the variance is large. Since it is intractable to estimate the true value of the gap $d$ during the course of optimization, we opt to design a **trust schedule** of maximally allowed gradient iterations that is correlated to the variance of $\mathbf{x}$ at each diffusion iteration $t$. In our experiments, we will demonstrate that simple schedules, such as a constant function $g_{\text{trust}}(t) = c$, or a linear function $g_{\text{trust}}(t) = m \cdot t + c$, work surprisingly well for the diverse set of tasks we attempted.

## 3.2 Early Termination Using State Manifold Boundaries

The previous section reformulates constrained guided diffusion as a gradient-based optimization, with our proposed algorithm designed to timely terminate the iterations based on the trustworthiness of the proxy objective. Equations 6 and 7 additionally require us to characterize the space that a forward sample $\mathbf{x}_t$ can possibly visit at diffusion level $t$, so that we can ensure, during inference time, that $\mathbf{x}'$ will not leave the state manifold where the base model is trained on. In practice, the robustness of diffusion models can produce valid samples even if the input is slightly outside of the state manifold, allowing the constraint $\mathbf{x}' \sim p(\mathbf{x}_t)$ to be relaxed. However, stepping outside of the state manifolds might require more unnecessary "corrective" steps, affecting the run-time performance. To speed up the computation during inference time, we describe a method for early termination of the optimization when the sample leaves the estimated boundary of the state manifold at each diffusion step.

We leverage the boundaries of a Diffusion model's intermediate state manifolds $\mathcal{M}_{t,\delta}$, which we define per diffusion timestep $t$ as the manifold on which a diffusion model has likely seen training data from with probability $\geq 1 - \delta$:

$$\mathcal{M}_{t,\delta} = \left\{ \mathbf{x}_t : \int q(\mathbf{x}_t|\mathbf{x}_0)p(\mathbf{x}_0)\mathrm{d}\mathbf{x}_0 \geq 1 - \delta \right\}. \tag{10}$$

Given sufficiently small $\delta$ and a sufficiently well-trained diffusion model, the idea is that any $\mathbf{x}_t \in \mathcal{M}_{t,\delta}$ will converge to some point $\mathbf{x}_0$ in the original data distribution $p(\mathbf{x}_0)$. As such, the optimization problem from Eq. 7 becomes:

$$\min_{\mathbf{x}'} L(\hat{\mathbf{x}}_0(\mathbf{x}'), \mathbf{y}) \quad \text{subject to } \mathbf{x}' \in \mathcal{M}_{t,\delta}, \quad |p(\mathbf{y}|\mathbf{x}') - p(\mathbf{y}|\hat{\mathbf{x}}_0)| < d. \tag{11}$$

By definition, $M_{t,\delta}$ is a larger manifold when $t$ is larger, meaning it gradually shrinks to true data manifold during diffusion inference. Nevertheless, $M_{t,\delta}$ would be challenging to compute in closed-form given the unknown true data distribution $p(\mathbf{x}_0)$. Our observation is that in all formulations of Diffusion models, we do have access to the model's predicted noise $\epsilon$. For a particular $\mathbf{x}'$, the ideal value for $\epsilon_\theta(\mathbf{x}', t)$ is

$$\epsilon_\theta(\mathbf{x}', t) = \int \frac{\mathbf{x}' - \sqrt{\alpha_t}\mathbf{x}_0}{\sqrt{1 - \alpha_t}}p(\mathbf{x}_0)\mathrm{d}\mathbf{x}_0 = \mathbb{E}_{\mathbf{x}_0}\left[\frac{\mathbf{x}' - \sqrt{\alpha_t}\mathbf{x}_0}{\sqrt{1 - \alpha_t}}\right]. \tag{12}$$

If $\mathbf{x}'$ is within the state manifold boundary, the integrand $\frac{\mathbf{x}' - \sqrt{\alpha_t}\mathbf{x}_0}{\sqrt{1-\alpha_t}}$ for each sample of $\mathbf{x}_0$ should correspond to a multivariant Gaussian $N(\mathbf{0}, \mathbf{I})$. This implies that we can estimate the boundary of $\mathcal{M}_{t,\delta}$ with $||\epsilon_\theta(\mathbf{x}', t)||$. When $||\epsilon_\theta(\mathbf{x}', t)||$ is far away from zero, $\frac{\mathbf{x}' - \sqrt{\alpha_t}\mathbf{x}_0}{\sqrt{1-\alpha_t}}$ is unlikely to be sampled from $N(\mathbf{0}, \mathbf{I})$. Consequently, $\mathbf{x}'$ is likely to be outside of the state manifold at the current diffusion step. In practice, we set such a threshold $\epsilon_{max}$ by observing the approximate average $||\epsilon_\theta(\mathbf{x}_t, t)||$ across several unconstrained samples running the base Diffusion model.

### 3.3 Algorithm

Comparing with standard DDIM sampling, our Trust Sampling algorithm (Algorithm 1) takes multiple gradient steps of constraint guidance up to a maximum of $J_t$, which denotes a max-iteration according to the trust schedule, $g_{\text{trust}}(t)$. We experiment with different linear schedules, detailed in the Experiments section, to show the positive impacts of our algorithm on the quality of generated data. The inner optimization loop will also be terminated by the condition when the magnitude of the predicted noise $\epsilon$ being larger than $\epsilon_{\max}$. Following Yang et al. [58], we normalize the gradient for numerical stability. $w$ is a constant step size which we keep either as 0.5 or 1.0 for each specific task.

---

**Algorithm 1:** Trust Sampling with DDIM

**Require:** $x_T \sim \mathcal{N}(\mathbf{0}, \mathbf{I})$, $T$, observation $y$, trust schedule $g_{\text{trust}}(t)$, norm upper bound $\epsilon_{\max}$, guidance weight $w$

1   **for** $t = T, \ldots, 1$ **do**
2     $\boldsymbol{\mu}_\theta \leftarrow \sqrt{\alpha_{t-1}}\hat{\boldsymbol{x}}_0(\boldsymbol{x}_t) + \sqrt{1 - \alpha_{t-1} - \sigma_t^2}\boldsymbol{\epsilon}_\theta(\boldsymbol{x}_t, t)$
3     $\boldsymbol{x}_{t-1}^*, j \leftarrow \boldsymbol{\mu}_\theta, 0$
4     $J_t \leftarrow g_{\text{trust}}(t)$
5     **while** $j < J_t$ **and** $||\boldsymbol{\epsilon}_\theta(\boldsymbol{x}_{t-1}^*, t)|| < \epsilon_{\max}$ **do**
6       $\boldsymbol{x}_{t-1}^* \leftarrow \boldsymbol{x}_{t-1}^* - w\nabla_{\boldsymbol{x}_{t-1}^*}L(\hat{\boldsymbol{x}}_0(\boldsymbol{x}_{t-1}^*), y)/||\nabla_{\boldsymbol{x}_{t-1}^*}L(\hat{\boldsymbol{x}}_0(\boldsymbol{x}_{t-1}^*), y)||$
7       $j \leftarrow j + 1$
8     **end**
9     $\boldsymbol{\epsilon}_t \sim \mathcal{N}(\mathbf{0}, \boldsymbol{I})$
10    $\boldsymbol{x}_{t-1} \leftarrow \boldsymbol{x}_{t-1}^* + \sigma_t\boldsymbol{\epsilon}_t$
11 **end**

---

**Adapting Inequality Constraints.** We use the mean-squared value over all constraint violations to compose $L$ in the case of equality constraints. However, we need to make an to handle inequality constraints. In the case of an inequality constraint $c_i(x) > a$, we choose formulate $L_i = \max(0, a - c_i(x))$. We then compose $L$ as the mean-squared value over all $L_i$.

# 4 Related Work

Our work is most closely related to zero-shot guided Diffusion methods for general loss functions. The seminal works of [11] and [24] introduced a method that alternates between taking one denoising step of the unconditional base diffusion model to maximize data distribution and taking one constraint gradient step to guide the model for conditional sample generation. This approach effectively balances data fidelity and conditional alignment. DSG [58] enhanced [11] by normalizing gradients in the constraint guidance term and implementing a step size schedule inspired by Spherical Gaussians. LGD-MC [47] addressed the inherent approximation errors in DPS by using multiple samples instead of a single point, which provided a better approximation of the guidance loss. Manifold Constrained Gradient (MCG) [10] and Manifold Preserving Guided Diffusion (MPGD) [19] use projections on the constraint gradient and predicted de-noised sample respectively to leverage the manifold hypothesis for better constraint following. In contrast, our work explores improving this paradigm using iterative gradient-based optimization.

Various methods have been developed specifically for guided diffusion of image restoration. RED-Diff [36] extends the principles of Regularization by Denoising (RED) for image noise removal [40] to a stochastic setting, offering a variational perspective on solving inverse problems with diffusion models. Techniques such as [10, 27, 55, 14, 49] assume linear distortion models and utilize the measurement operator matrix to improve guidance for image restoration. To handle non-linear distortion models, approaches like [41] and [56] have been proposed. These methods can accommodate complex distortion but require specialized initialization schemes, which limits their general applicability. In contrast, our approach initializes from the standard unit Gaussian, ensuring broader applicability in general tasks. Similarly, ΠGDM [46] addresses inverse problems for image restoration with diffusion, but it is confined to certain loss types, while RePaint [31] enhances diffusion-based image in-painting by repeating crucial diffusion steps to improve fidelity.

Recent advancements like [59] and [6] tackle guided diffusion tasks based on conditional models, with conditions including textual information. FreeDOM [59] additionally adopts an energy-based framework and generalizes the repeating strategy found in RePaint[31] with a novel time travel strategy. DiffPIR [62] balances the data prior term from the unconditional diffusion model with the constraint term from measurement loss to improve image restoration tasks.

Other methods adopt additional training for controlled diffusion. Ambient Diffusion Posterior Sampling [1] builds upon DPS [11] by training the base model on linearly corrupted data. [48] learns a score function for the noise distribution, specifically targeting structured noise in images. ControlNet [60] and OmniControl [57] train additional Diffusion branches to process input constraints and conditions, achieving notable results in image or motion domains. DreamBooth [42] fine-tunes a base diffusion model to place subjects in different backgrounds using a few images, demonstrating versatility in content generation. Other notable related works include [15], which focuses on composing multiple diffusion models. The proposed MCMC framework replaces simple gradient addition with a more robust iterative optimization process, similar to our framework for solving guided diffusion. D-PNP [17] reformulates diffusion as a prior for various guidance tasks but has been observed to struggle with more complex diffusion models, such as those trained on ImageNet [12].

# 5 Experiments

We evaluate our method on two drastically different domains: images and 3D human motion. In both domains, we compare against recent zero-shot guided diffusion algorithms for solving general constraint diffusion: DPS [11], DSG [58], and LGD-MC [47].

## 5.1 Image Experiments

**Tasks.** We evaluate our method on three challenging image restoration problems: Super-resolution, Box Inpainting, and Gaussian Deblurring. These common linear inverse problems are standard across DPS [11], DSG [58], and LGD-MC [47]; we note that in this paper, we do not inject noise into the initial observations. Each of these image restoration problems can be thought of as a constraint satisfaction problem, where the constraint is that the generated picture appear the same as the source image upon applying the particular distortion. Distortion for these problems, respectively, was performed via (i) bicubic downsampling by $4\times$, (ii) randomly masking a $128 \times 128$ square region

|  | SR ($\times 4$) | | Inpaint (box) | | Deblur (Gauss) | |
|---|---|---|---|---|---|---|
| Methods | FID ↓ | LPIPS ↓ | FID ↓ | LPIPS ↓ | FID ↓ | LPIPS ↓ |
| DPS | 29.48 | 0.212 | 20.19 | 0.140 | 23.59 | 0.195 |
| DPS+DSG | 27.06 | 0.193 | 18.92 | **0.137** | 24.06 | 0.194 |
| LGD-MC ($n=10$) | 29.59 | 0.212 | 20.15 | 0.141 | 27.38 | 0.229 |
| LGD-MC ($n=100$) | 29.54 | 0.212 | 20.13 | 0.140 | 27.23 | 0.228 |
| Trust (ours) | **16.99** | **0.156** | **15.28** | 0.141 | **21.19** | **0.176** |

Table 1: Quantitative evaluation (FID, LPIPS) of solving linear inverse problems on 1000 validation images of FFHQ $256 \times 256$. **Bold**: best, red: worst.

|  | SR ($\times 4$) | | Inpaint (box) | | Deblur (Gauss) | |
|---|---|---|---|---|---|---|
| Methods | FID ↓ | LPIPS ↓ | FID ↓ | LPIPS ↓ | FID ↓ | LPIPS ↓ |
| DPS | 111.53 | 0.353 | 142.03 | 0.282 | 152.57 | 0.442 |
| DPS+DSG | 148.53 | 0.438 | 115.90 | 0.247 | 145.64 | 0.406 |
| LGD-MC ($n=10$) | 110.36 | 0.353 | 142.77 | 0.280 | 142.33 | 0.424 |
| LGD-MC ($n=100$) | 108.00 | 0.349 | 131.23 | 0.282 | 152.53 | 0.444 |
| Trust (ours) | **55.24** | **0.236** | **99.87** | **0.210** | **69.49** | **0.266** |

Table 2: Quantitative evaluation (FID, LPIPS) of solving linear inverse problems on 100 validation images of ImageNet $256 \times 256$. **Bold**: best, red: worst.

(sampled uniformly within a 16 pixel margin of each side), and (iii) Gaussian blur kernel of size $61 \times 61$ with standard deviation 3.0. We experimented on two datasets: FFHQ $256 \times 256$ [26] and ImageNet $256 \times 256$ [12] on 100 validation images each given our limited compute access. For a fair comparison between methods, we used the same pretrained unconditional diffusion models across methods for FFHQ [11] and ImageNet [13] following previous works. Quantitative evaluation of images is performed with two widely used metrics for image perception: Fréchet Inception Distance (FID) [20] and Learned Perceptual Image Patch Similarity (LPIPS) [61].

**Results.** Quantitative evaluation results can be seen in Tables 1 and 2. Our method outperforms diffusion model baselines by a significant margin across all three image tasks on both FID and LPIPS and on both FFHQ and ImageNet. Qualitative results can be seen in Fig. 2. In super-resolution, Trust Sampling shows an ability to adhere to the original down-sampled image better, even recovering text much better. In box inpainting, Trust Sampling fills in the box with realistic output; for example, the eyes in the human faces generated on the right of Fig. 2 are much more natural.

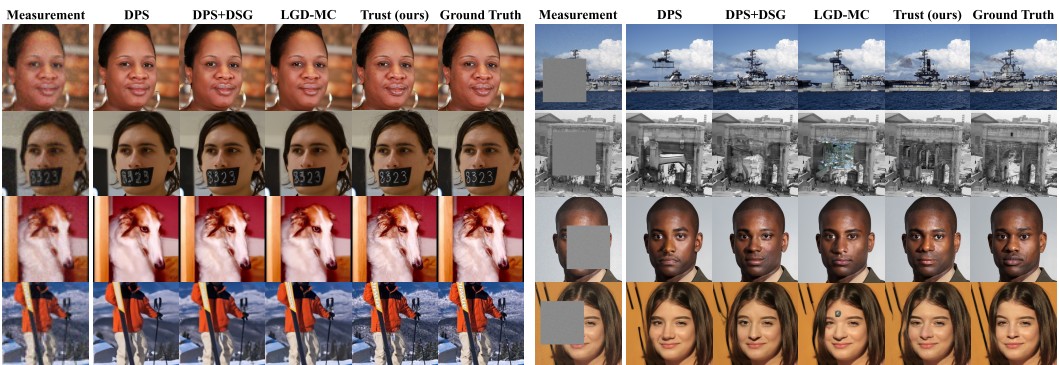

Figure 2: Results on solving linear inverse problems. The left shows examples of box inpainting; the right shows examples of super-resolution.

| Methods | root tracking | | | right hand & left foot tracking | | |
|---|---|---|---|---|---|---|
| | FID ↓ | Diversity → | Const. [m] ↓ | FID ↓ | Diversity → | Const. [m] ↓ |
| DPS | **542.8** | 23.8 | 0.13 | **604.7** | 22.5 | 0.12 |
| DPS+DSG | 715.1 | 25.0 | 0.022 | 865.5 | 24.0 | **0.035** |
| LGD-MC ($n = 10$) | 578.6 | 21.6 | 0.031 | 715.2 | 22.6 | 0.056 |
| LGD-MC ($n = 100$) | 579.3 | 22.6 | **0.006** | 731.6 | 23.1 | 0.052 |
| Trust (ours) | 561.6 | **21.5** | 0.026 | 694.1 | **20.4** | 0.038 |
| GT | - | 17.3 | - | - | 17.3 | - |

Table 3: Evaluation of FID, Diversity, and Constraint Violation in meters for motion tasks: root tracking and right hand & left foot tracking. **Bold**: best, red: worst. Computational budget for all methods is 1000 NFEs.

## 5.2 Human Motion Experiments

**Unconditional Motion Diffusion Model.** For all tasks we use the same unconditional diffusion model, which we trained on the AMASS [33, 2, 28, 16, 8, 7, 30, 37, 34, 29, 53, 25, 51, 44, 3] dataset excluding the following datasets that are used for testing: danceDB [5], HUMAN4D [9] and Weizmann [35]. The architecture is an adapted version of the EDGE motion model [52], where we removed the branches handling conditions.

**Metrics.** We evaluate DPS, DPS+DSG, LGD-MC, and Trust on several constrained motion generation tasks. We train an autoencoder and use the encoder as a feature extractor for motion clips, to allow for calculation of motion realism and diversity metrics [38]. We use the following metrics to evaluate performance:

- FID: We extract features using the aforementioned encoder and calculate FID between different methods vs. ground-truth, as in Action2Motion [18].
- Diversity: We extract features and calculate the diversity metric as in Action2Motion [18] for the generated and ground truth motions. A result is claimed better than others if its score is closer to the score of the ground truth.
- Constraint Violation: A task-dependent metric that describes how well the generated motion adheres to the provided constraints.

**Tasks.** We first evaluate on two tasks where we have ground truth motions from the test dataset: *root trajectory tracking* and *right hand & left foot trajectory tracking*. Here the diffusion model should be guided to generate natural human movements that closely follow specified root motions or hand/foot motions. Note that the generations do *not* need to match the ground-truth motions due to under-specification of the constraints; we are only using them for generating the control constraints which are guaranteed to be physically feasible for human movements. Specifically, we randomly select a total of 1000 slices from the mentioned three test sets, and we extract their root motion and right hand and left ankle motion as constraint signals for the respective tasks. Note also that the observation mapping, from full motion states to the constraint signals, is highly non-linear in the hand/foot tracking task. This is because the full motion state of Diffusion only uses local joint rotations, but the hand/foot trajectory is defined in the global Cartesian space (see EDGE [52] for more details).

**Results.** Our method strikes the best balance to matching the constraint without sacrificing realism nor diversity. DPS has the best FID score closely followed by ours. However, this comes at a large cost for DPS that violates the constraints. DSG satisfies constraints slightly better than our method, but it sacrifices both diversity and realism significantly. Our method outperforms DPS and DSG on the Diversity score for both *root tracking* and *right hand & left foot tracking*. While LGD-MC balances fidelity and constraint following better, it still has worse fidelity than Trust and struggles with harder tasks such as *right hand & left foot tracking*. Note that for these tasks the constraint metric is the root-mean-square tracking error in meter. The difference between ours, DSG, and LGD-MC are hardly noticeable when performing visual comparison between the generated motions, especially for *root tracking*. Visualizations that support these observations are in Appendix D, Fig. 6, but are best viewed in the **supplementary video**.

**More Challenging Tasks.** We further experimented our method with more difficult tasks such as sparse spatio-temporal constraints, inequality and highly non-linear constraints, and compositing multiple constraints. This is in drastic contrast to the image tasks, where a single "dense" (closer to being fully-specified) constraint must be satisfied. We designed the following tasks with additional two composite constraints on each—a translation constraint on the initial and the final frames.

- *Obstacle Avoidance*: We add an inequality constraint to avoid penetration between any joint and three pseudo-randomly placed obstacle spheres.
- *Jump*: We add an inequality constraint at the middle frame, to impose that all joints have a vertical position that is higher than a selected value between 0.6 m and 1.0 m.
- *Angular Momentum*: We add an inequality constraint to impose different minimum values for the average angular momentum around a horizontal axis. This serves as a way to control dynamicism of a motion. Angular momentum is approximated as: $\sum_{i=1}^{4} \boldsymbol{v}_i \times \boldsymbol{p}_i$. with $\boldsymbol{v}_i, \boldsymbol{p}_i$ the relative velocity and position of an end effector (wrists and ankles) with respect to the root.

As quantitative metrics would not be informative in these tasks (for example, it is not reasonable to compute the distributional distance between ground-truth test set and jumping motions), we focus on qualitative demonstrations. We consistently found that for easier inequality constraints (e.g. lower jumping heights) all methods could match the constraints. Howver, our method was more robust when constraints became harder, while DSG sacrificed physical realism and DPS violated the constraints. See Fig. 6, and **supplemental videos** for more details on these observations.

### 5.3 Ablations

To examine the influence of Trust sampling, we performed ablations on the same three image tasks on FFHQ. In addition to FID and LPIPS, we look at the number of neural function evaluations (NFEs) as an implementation-agnostic metric of efficiency. In our case, NFEs is the number of times a pass through the pretrained model occurs.

**Trust Scheduling.** We decouple just the trust schedule and do not use state manifold estimates for this part. The results (Table 4) show that our method is not sensitive to scheduling parameters as all schedules still outperform the DPS and DSG baselines on all three image tasks by significant margins. Within the different schedules, we see that linear schedules with non-zero slope (i.e. non-constant schedules) typically outperform constant schedules. This aligns with our notion of trust, as earlier diffusion steps tend to be noiser and therefore the proxy constraint function is less trustworthy, so it is less productive to take gradient steps at earlier times. Although linear trust schedule is better than constant schedules, the results indicate the best slope is dependent of the task and NFEs.

**Fewer NFEs.** Table 4 also shows when decreasing NFEs from 1000 (same as baselines) to 600, the performance of our method barely drops and are still significantly better than baselines. To control the desired number of NFEs (1000 or 600 in our experiments), we choose a few combinations of the slope $m$ and the offset $c$ of the trust schedule $g_{\text{trust}}(t) = m \cdot t + c$, such that $\sum_{t=1}^{T} g_{\text{trust}}(t)$ equals the desired number of NFEs, where $T$ is the total number of diffusion iterations.

**Manifold Boundary Estimates.** We examined the effect of using manifold boundary estimates on the image tasks on FFHQ and ImageNet. We compare the effect of manifold boundary estimates when added to the trust schedule, as compared to only trust scheduling. Table 5 shows the results of using manifold boundary estimates. The use of manifold boundary reduces the needed NFEs by 10–20% without any substantial loss in quality, resulting in better compute efficiency. This performance boost is evidently robust across image task, dataset, and NFEs. Table 5 also shows that if instead of adopting manifold boundary, we want to achieve the same NFE save by tuning the start and end points of the linear schedule, model quality can suffer. Table 6 shows the effect of varying $\epsilon_{\max}$. We observe that $\epsilon_{\max}$ generally has an acceptable range (e.g. 440-442 for FFHQ Super Resolution (4×)), within which performance varies only slightly. For the motion tasks we did not find a significant effect when introducing manifold boundary estimates.

| parameters | | | SR ($\times 4$) | | Inpaint (box) | | Deblur (Gauss) | |
|---|---|---|---|---|---|---|---|---|
| Total NFEs | Start | End | FID ↓ | LPIPS ↓ | FID ↓ | LPIPS ↓ | FID ↓ | LPIPS ↓ |
| 1000 | 4 | 4 | 36.95 | **0.150** | 34.72 | 0.146 | 47.25 | 0.179 |
| 1000 | 2 | 6 | **35.73** | 0.152 | **32.63** | **0.145** | 45.56 | **0.173** |
| 1000 | 0 | 8 | 36.26 | 0.156 | 35.08 | 0.148 | **42.98** | 0.173 |
| 600 | 2 | 2 | 45.01 | 0.153 | 44.22 | 0.178 | 57.12 | 0.199 |
| 600 | 1 | 3 | 41.56 | 0.159 | 44.81 | 0.178 | 54.11 | 0.195 |
| 600 | 0 | 4 | **34.94** | **0.149** | **38.70** | **0.151** | **48.94** | **0.181** |
| baseline (DPS) | | | 64.66 | 0.230 | 51.25 | 0.176 | 60.91 | 0.226 |
| baseline (DSG) | | | 60.23 | 0.214 | 58.30 | 0.179 | 59.59 | 0.212 |

Table 4: Trust scheduling ablation study on NFEs and different trust schedules. Metrics calculated on linear inverse problems on 100 validation images of FFHQ $256 \times 256$. "Start" and "End" indicate the boundary conditions of the trust schedule: $g_{\text{trust}}(1) =$ Start and $g_{\text{trust}}(T) =$ End. **Bold**: best among same NFEs, underline: second best among same NFEs.

| parameters | | SR ($\times 4$) | | | Inpaint (box) | | | Deblur (Gauss) | | |
|---|---|---|---|---|---|---|---|---|---|---|
| Bound | Start, End | NFEs | FID ↓ | LPIPS ↓ | NFEs | FID ↓ | LPIPS ↓ | NFEs | FID ↓ | LPIPS ↓ |
| ✗ | 0, 3 | 500 | 42.31 | 0.160 | **500** | 48.85 | 0.186 | 500 | 56.57 | 0.195 |
| ✗ | 1, 3 | 600 | 41.56 | 0.159 | 600 | 44.81 | 0.178 | 600 | 54.11 | 0.195 |
| ✗ | 0, 4 | 600 | **34.94** | **0.149** | 600 | **38.70** | **0.161** | 600 | **48.94** | 0.181 |
| ✓ | 0, 4 | **497** | 37.52 | 0.150 | 561 | 41.87 | 0.169 | **498** | 49.95 | **0.181** |

Table 5: FFHQ Manifold boundary ablations. Metrics calculated on linear inverse problems on 100 validation images of FFHQ $256 \times 256$. **Bold**: best, underline: second best.

| parameters | | SR ($\times 4$) | |
|---|---|---|---|
| $\epsilon_{\max}$ | Start, End | FID ↓ | LPIPS ↓ |
| 438.0 | 15, 15 | 50.81 | 0.191 |
| 439.0 | 10, 10 | 40.90 | 0.165 |
| 440.0 | 5, 5 | 35.85 | 0.153 |
| 441.0 | 4, 4 | **35.46** | **0.149** |
| 442.0 | 4, 4 | 36.06 | 0.150 |

Table 6: $\epsilon_{\max}$ ablations. Metrics calculated on Super Resolution ($4\times$) on 100 validation images of FFHQ $256 \times 256$. To isolate purely the effect of $\epsilon_{\max}$ while keeping the number of NFEs comparable, constant linear schedules were chosen so that the number of NFEs was close to 1,000. **Bold**: best.

## 6 Conclusion

We introduce trust sampling, a novel and effective method for guided diffusion, addressing the current limitations of meeting challenging constraints. By framing each diffusion step as an independent optimization problem with principled trust schedules, our approach ensures higher fidelity across diverse tasks. Extensive experiments in image super-resolution, inpainting, deblurring, and various human motion control tasks demonstrate the superior generation quality achieved by our method.

Our findings indicate that trust sampling not only enhances performance but also offers a flexible and generalizable framework for future advancements in constrained diffusion-based modeling. To further improve generation quality, future research should adopt a holistic approach by incorporating additional concepts from traditional numerical optimization into this framework, beyond just the termination criterion. This includes techniques such as step size line search and fast approximation of higher-order derivatives. Moreover, automating the setting of heuristic parameters, which are currently manually adjusted for each base diffusion model, would be beneficial.

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

# A   Experiment Details

**Image Parameters.**   The parameters used to for all tasks can be found in Table 7. In implementing linear schedules, we found the most effective class of trust schedule to be *stochastic linear schedules*, where the expected values of iteration limits over diffusion time, $\mathbb{E}[J_t]$, form an arithmetic sequence, and the integer iteration limit $J_t$ is determined at runtime by randomly rounding up with probability $\mathbb{E}[J_t] - \lfloor \mathbb{E}[J_t] \rfloor$.

| Max NFEs | DDIM Steps | Dataset | Task | Start | End | $\epsilon_{\max}$ |
|---|---|---|---|---|---|---|
| 1000 | 200 | FFHQ | SR | 2 | 6 | 441 |
| 1000 | 200 | FFHQ | Inpaint | 2 | 6 | 442 |
| 1000 | 200 | FFHQ | Deblur | 2 | 6 | 441 |
| 1000 | 200 | ImageNet | SR | 0 | 8 | 441 |
| 1000 | 200 | ImageNet | Inpaint | 0 | 8 | 442 |
| 1000 | 200 | ImageNet | Deblur | 0 | 8 | 441 |
| 600 | 200 | FFHQ | SR | 0 | 4 | 441 |
| 600 | 200 | FFHQ | Inpaint | 0 | 4 | 442 |
| 600 | 200 | FFHQ | Deblur | 0 | 4 | 441 |

Table 7: Parameters used for all experiments. Start and end refer to the start and end of the stochastic linear trust schedules.

**Motion Parameters.**   For all motion experiments, we match the computational budget (NFEs) between methods: we use 1000 DDIM steps for DPS and DPS+DSG. We spend between 950 and 1000 NFEs for Trust by using 200 DDIM steps with a stochastic stochastic linear schedule using Start 0 and End 8. As mentioned in the experiments, we did not find a significant effect when introducing manifold boundary estimates for motion and thus there is no $\epsilon_{\max}$ set for the motion experiments.

# B   Compute Resources

For image tasks, we used pretrained models for FFHQ and ImageNet. We ran inference on an A5000 GPU, which takes roughly 1 minute to generate an image for FFHQ and 6 minutes to generate an image for ImageNet, due to the larger network size. For motion tasks, the diffusion model was trained on a single A4000 GPU for approximately 24 hours. Inference does not require a large GPU and generating a single motion trial, without batching, takes less than a 30s.

# C   Qualitative Samples for Images

Figures 3, 4, and 5 illustrate several examples of Trust sampling on Gaussian Deblurring, Box Inpainting, and Super-Resolution respectively on both the FFHQ and ImageNet datasets.

# D   Qualitative Samples for Motion

Fig. 6 illustrates several examples of complex motions generated by trust sampling. More results are presented in the **Supplemental Video**.

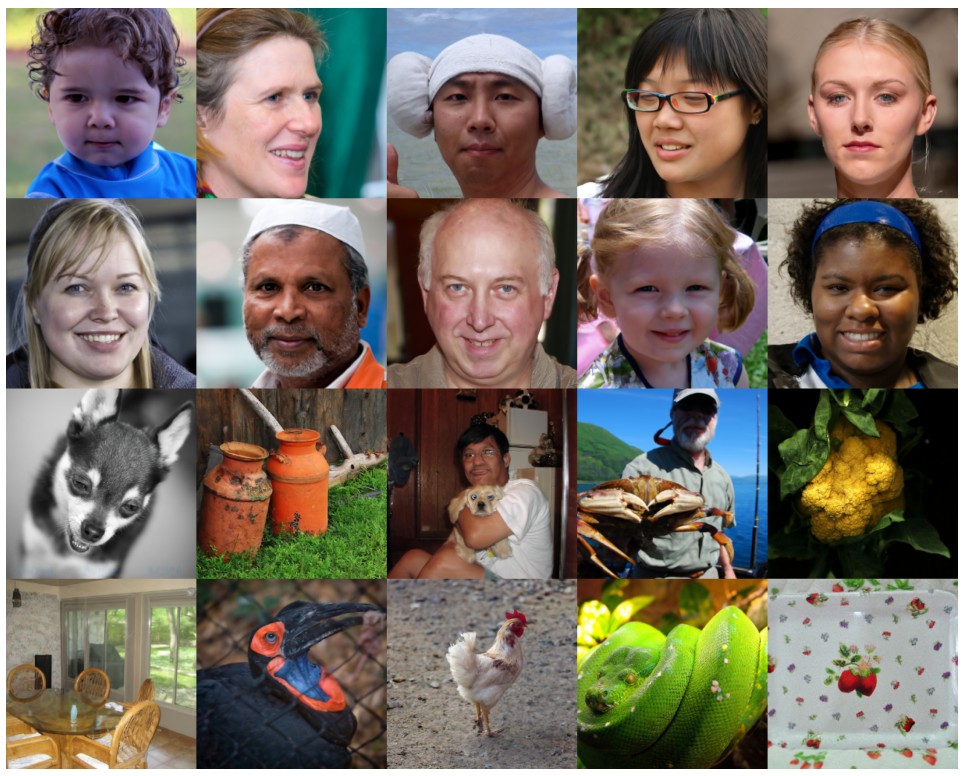

Figure 3: Qualitative results for Trust on Gaussian Deblurring. The first two rows of images are from FFHQ, and the latter two rows of images are from ImageNet.

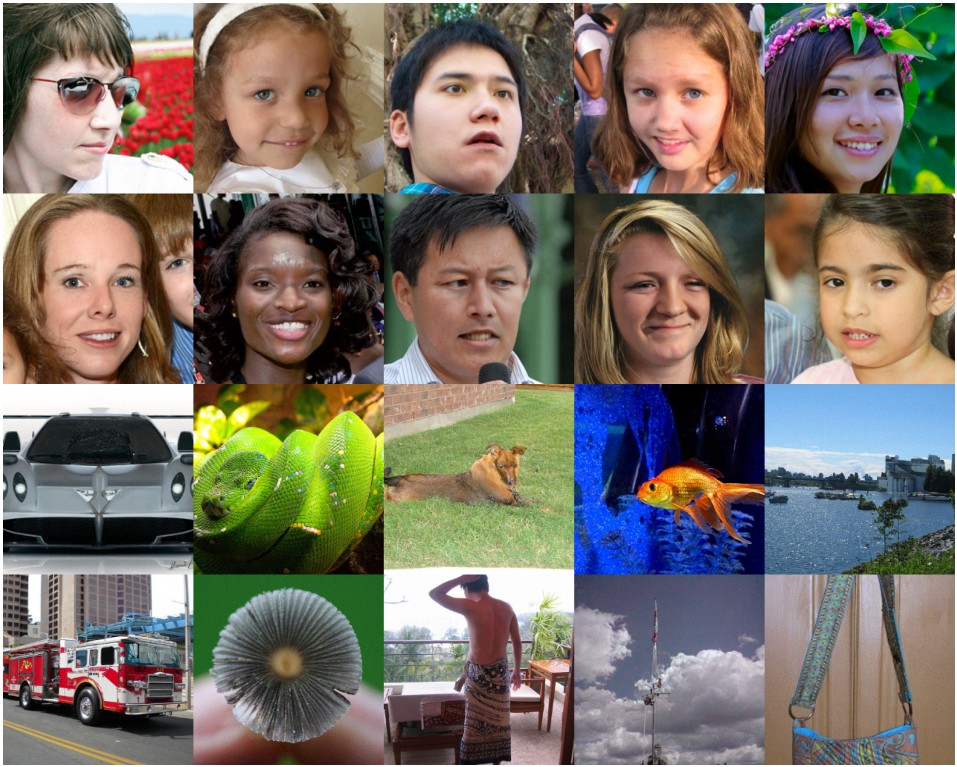

Figure 4: Qualitative results for Trust on Box Inpainting. The first two rows of images are from FFHQ, and the latter two rows of images are from ImageNet.

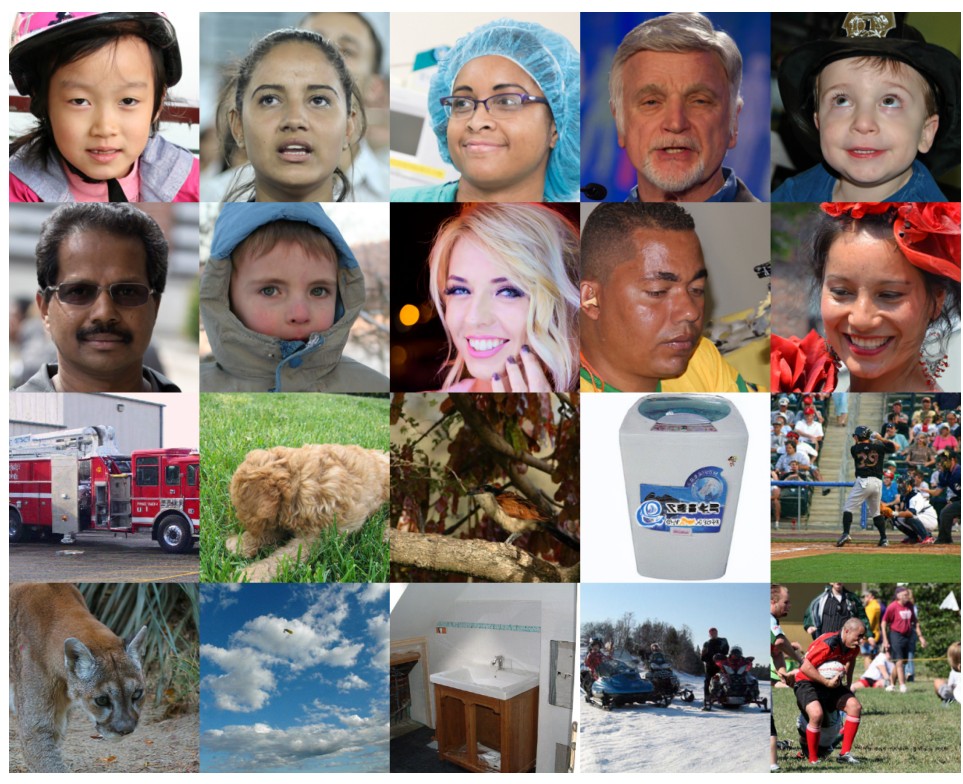

Figure 5: Qualitative results for Trust on Super-Resolution. The first two rows of images are from FFHQ, and the latter two rows of images are from ImageNet.

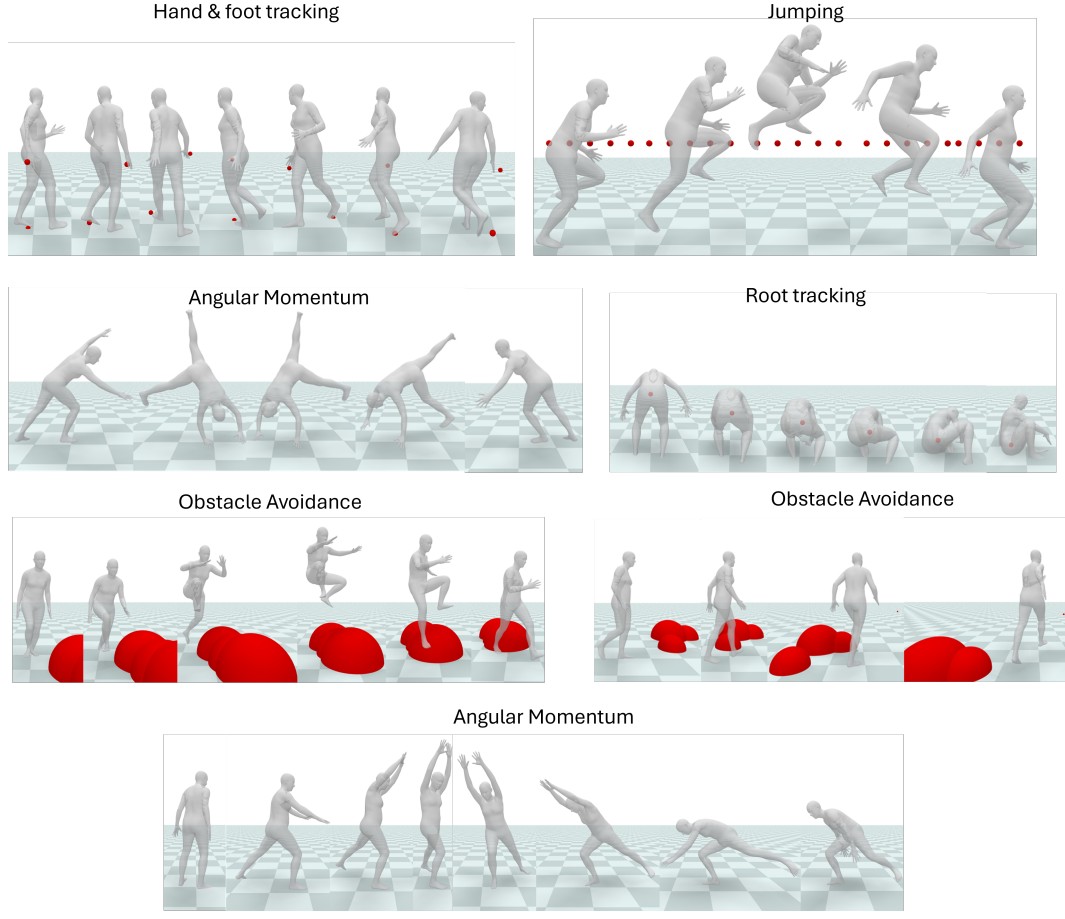

Figure 6: Qualitative results for Trust on different motion tasks. For "Jumping" the horizontal dotted line indicates the required height to be cleared.

