# OpenReview forum: "Constrained Diffusion with Trust Sampling"
_NeurIPS.cc/2024/Conference — NeurIPS 2024 poster_

### Official Review · Reviewer_Pivm · 2024-07-01

**Soundness:** 2
**Presentation:** 3
**Contribution:** 2
**Rating:** 5
**Confidence:** 4

**Summary:**

This paper addresses this limitation by rethinking diffusion without training loss guidance from an optimization perspective. They formulate a series of constrained optimizations throughout the inference process of the diffusion model. In each optimization, they allow the sample to take multiple steps along the gradient of the surrogate constraint function. The termination conditions come from two aspects: one is the accuracy of the approximate surrogate, and the other is the estimation of the manifold.

**Strengths:**

1. The motivation for this paper seems reasonable to me, and they believe that the one-step guidance of DPS is suboptimal.

2. The proposed direction has improved the performance compared to the two baselines DPS and DSG.

**Weaknesses:**

1. First of all, I have doubts about the theory of this article. Equation 12 in the article seems to be written incorrectly. As far as I know, $\epsilon_\theta(\mathbf{x}',t)=\int(\mathbf{x}'-\sqrt{\alpha_t}\mathbf{x_0})/(\sqrt{1-\alpha_t})p(\mathbf{x}_0|\mathbf{x}')\mathrm{d}\mathbf{x}_0$, it should be conditional expectation here.

2. The motivation for this part is also not very clear. Why does the above integral correspond to a multivariate Gaussian (Line 156)?

3. The number of test samples used in the image experiment is a bit small. Why didn't the author use a data set similar to DPS?

4. In addition, the proposed method still seems to require more than 500 NFE, similar to DPS, and I am a little worried about its efficiency and the need for improvement.

5. DPS itself requires careful tuning of hyperparameters, such as the guidance rate. The proposed method introduces new hyperparameters $m·t+c$ and $\epsilon_max$, which raises scalability concerns.

**Questions:**

see Weaknesses.

**Limitations:**

The authors discuss the limitations of their work.

---

> ### Author Rebuttal · Authors · 2024-08-07
>
> We thank the reviewer for acknowledging the performance of our method and raising valuable questions for discussion. Here are our responses to your questions:
> 1. Re: conditional probability in Eq. (12). The reviewer is correct that equation 12 should indeed take the form of $p(x_0|x’)$ instead of $p(x_0)$. We apologize for this careless error and this should not affect the following reasoning.
> 2. Re: explain L156-157 better: We see where the confusion comes from and we will revise the text to make it clearer. We are saying part of the *integrand* should be a sample from unit Gaussian, per definition of the forward noising process of Diffusion, not the *integral*. Specifically, we are saying that if we assume $\mathbf x_0$ to be the true original sample drawn from the data distribution, the portion of the integrand $(\mathbf x' - \sqrt{\alpha_t} \mathbf x_0)/\sqrt{1-\alpha_t}$ provides us the original noise $\boldsymbol{\epsilon}$ that was drawn from a multivariate normal distribution $N(\mathbf 0, \mathbf I)$. However, since we're not sure which $\mathbf x_0$ is the correct one, we perform a "weighted sum" (an integral) of $\boldsymbol{\epsilon}$ over all possible $\mathbf x_0$ starting points, in which each term $\boldsymbol{\epsilon}$ is a sample of a normal distribution. Of course, these samples from normal distributions are not independent, but because each term is sampled from a normal distribution, each individual term is unlikely to have a high norm. Therefore, for the overall integral to have a high norm, the individual terms must be aligned along a particular direction. This indicates that a high $||\boldsymbol{\epsilon}_\theta ||$ means an out of distribution $\mathbf x'$.
> 3. Re: same test set size as DPS: Unfortunately, reproducing DPS with the same test set size of 1,000 images would take around 20 hours for FFHQ and 100 hours for ImageNet for each configuration of parameters. Note that our paper includes many more quantitative ablation studies than the DPS paper. That said, we will update all numbers in our tables with the DPS test set size in our final version. This update should not affect the overall narrative due to the significant quality gap between ours and the baselines.
> 4. Re: efficiency: We agree there can be further room for improvement in terms of efficiency, though the focus of our evaluation is to showcase that with a similar level of NFEs, our method gives higher-quality results (as shown by quantitative metrics and qualitative comparisons) than existing SoTA methods.
> 5. Re: the introduction of two additional parameters. While our method has additional parameters, the robustness of our parameter space is very different. DPS indicates high sensitivity to the hyperparameters such as guidance rate (also pointed out by the LGD-MC paper), and DPS+DSG is very sensitive to several hyperparameters such as their guidance rate and interval. On the other hand, trust sampling isn’t nearly as sensitive: Tables 4 and 5 indicate a wide range of parameter choices for our trust schedules, most of which perform better than baselines. Furthermore, our new table (Table 4 of our one page PDF) shows $\epsilon_\mathrm{max}$ has a wide range of applicability as well. Therefore, despite there being more parameters, our method is rather robust with respect to our parameters.

---

> > ### Author Response · Authors · 2024-08-13
> >
> > Hello! With the rebuttal discussion period ending soon, we would like to kindly ask if our rebuttal addresses your questions, and if you have additional questions or concerns. Thank you for your time and effort on this review!

---

> ### Comment · Reviewer_Pivm · 2024-08-13
> **Response to author**
>
> Thanks to the author for the great effort put into the rebuttal, which has alleviated some of my concerns.
>
> However, I found some work that also constrains DPS [1] during the rebuttal, which seems to be highly relevant to this paper, as it aim to constrain diffusion on a specified manifold. The author seems to have ignored these developments, which may need further discussion.
>
> [1] MANIFOLD PRESERVING GUIDED DIFFUSION, ICLR 2024.

---

> ### Author Response · Authors · 2024-08-13
>
> We thank the reviewer for their continued effort in helping us improve this manuscript. We are pleased that our efforts to address the previous concerns have been acknowledged.
>
> Regarding the new issue raised, we apologize for the omission of the related work [1]. However, after carefully reviewing we believe that our paper's contributions remain distinct. The “key idea” of [1] is that “the information bottleneck in autoencoders naturally incorporates the manifold hypothesis.” While this approach is relevant for many modern image diffusion networks, which often utilize autoencoders for dimensionality reduction, it does not apply as readily to other domains, such as 3D human motion we experimented with in this paper. Indeed, [1] is only tested on image models.
>
> Our method, like DPS, DSG, and LGD-MC, is designed to be domain-agnostic. While [1] proposes a weaker variant that could be applied across domains—by changing \nabla_x_t to \nabla_x_0 in the standard loss guidance equation and using the gradient to update x_0 instead of x_t (note that the loss L is always computed on x0 regardless)—this is a commonly known implementation choice among researchers in the field, for example mentioned in the first two paragraphs of Section 4.2 of https://arxiv.org/pdf/2305.12577. We also experimented with this variant early in our project but did not observe any noticeable improvements for our test problems, possibly also due to the potential issues outlined in the aforementioned paper.
>
> We want to assure all reviewers that this omission was an honest mistake and not a deliberate oversight. We made every effort to include relevant papers by utilizing common online tools. However, when we reviewed the papers citing the seminal work DPS and the state-of-the-art paper LGD-MC, Google Scholar did not list [1]. Only now we could realize that this may have been due to a technical issue with Google Scholar not indexing the paper correctly: https://scholar.google.com/scholar?hl=en&as_sdt=2005&sciodt=0%2C5&cites=696239910969416231&scipsc=1&q=manifold+preserving&oq=
>
> In summary, we believe that the added value of our manuscript to the research community remains intact, with or without [1]. Our work provides new theoretical insights, introduces a novel method which can potentially be combined with [1] for image domain problems, and presents a set of new experimental results.
>
> We will include a more thorough discussion and comparison with this paper in the next revision.

---

### Official Review · Reviewer_bnFv · 2024-07-07

**Soundness:** 3
**Presentation:** 4
**Contribution:** 3
**Rating:** 5
**Confidence:** 5

**Summary:**

The paper presents a method to enhance training-free loss-guided diffusion sampling. The key contributions are:

1. Introduction of Trust Sampling: A novel method called Trust Sampling is proposed, which diverges from the traditional approach of alternating between diffusion steps and loss-guided gradient steps. Instead, it treats each timestep as an independent optimization problem, allowing multiple gradient steps on a proxy constraint function at each diffusion step.

2. Early Termination and State Manifold Estimation: The method includes a mechanism to estimate the state manifold of the diffusion model, enabling early termination if the sample starts to deviate significantly from the expected state. This ensures the proxy constraint function remains trustworthy.

3. Optimization Perspective: The paper reformulates training-free guided diffusion as a constrained optimization problem, providing more flexibility and robustness across various domains and tasks. This approach addresses the limitations of previous methods, such as sensitivity to initialization and performance degradation with fewer neural function evaluations.

4. Experimental Validation: The efficacy of Trust Sampling is demonstrated through extensive experiments in different domains, including image and 3D motion generation. The method shows significant improvements in generation quality and constraint satisfaction compared to existing techniques.

5. Generalization Across Tasks: The paper demonstrates the generality of Trust Sampling across various image tasks (e.g., super-resolution, inpainting, gaussian deblurring) and motion tasks (e.g., trajectory tracking, obstacle avoidance), highlighting its versatility and effectiveness.

**Strengths:**

1. The paper proposes Trust Sampling, a method that diverges from traditional guided diffusion approaches. Instead of alternating between diffusion and gradient steps, Trust Sampling treats each timestep as an independent optimization problem. This innovation provides a new perspective on training-free guided diffusion, addressing some of the key limitations of existing methods.
2. The quality of the paper is reflected in the thoroughness of its methodology and the robustness of its experimental validation.
3. The paper is well-written and clearly structured, making it accessible to both experts and those new to the field.
4. The significance of the paper lies in its potential to impact a wide range of applications in generative modeling.

**Weaknesses:**

1. The Trust Sampling algorithm, which takes multiple gradient steps of constraint guidance on each predicted $x_{t}$, is similar to the  corrector stage of PC sampling [1].
2. Training-free loss-guided diffusion sampling have been applied not only on inverse problem but also on variable tasks [2-6], such as refined text-to-image and layout-to-image. The authors need to further verify the effectiveness of Trust Sampling on these tasks.

[1] Score-based Generative Modeling through Stochastic Differential Equations

[2] Counting Guidance for High Fidelity Text-to-Image Synthesis

[3] Fine-grained Text-to-Image Synthesis with Semantic Refinement

[4] Attend-and-Excite: Attention-Based Semantic Guidance for Text-to-Image Diffusion Models

[5] BoxDiff: Text-to-Image Synthesis with Training-Free Box-Constrained Diffusion

[6] LoRA-Composer: Leveraging Low-Rank Adaptation for Multi-Concept Customization in Training-Free Diffusion Models

**Questions:**

Please see weaknesses.

**Limitations:**

The authors have adequately described the limitations and potential negative societal impact of their work.

---

> ### Author Rebuttal · Authors · 2024-08-07
>
> We thank the reviewer for acknowledging the novelty of our method, technical quality, structure of our paper, and significance of our work, and raising valuable questions for discussion.
>
> Re: more evaluation tasks: the reviewer suggested several good tasks to further test our method: for example [2] and [4] use neural-network predicted values instead of analytical constraint functions to compute the guidance loss during Diffusion, which can fit into our framework. We will strive to include additional tasks in the final version of the paper.

---

> > ### Comment · Reviewer_bnFv · 2024-08-11
> >
> > I acknowledge having read the authors' rebuttal. My overall assessment of the paper remains unchanged, and I continue to support my current rating.

---

### Official Review · Reviewer_rJ33 · 2024-07-12

**Soundness:** 2
**Presentation:** 3
**Contribution:** 2
**Rating:** 5
**Confidence:** 3

**Summary:**

This paper proposes a trust sampling scheme which incorporates given constraints loss function as guidance for constrained generation. This approach conducts early stop while detecting a mismatch between the predicted noises magnitude of sample and the noisy level at the current state manifold at each diffusion step. In experiment section, this approached is applied to image generation and human motion tasks both across various applications.

**Strengths:**

1. This approach can deal with given constraints using the pre-trained unconditional diffusion models without more training.
2. Paper is easy to follow.

**Weaknesses:**

1. It seems that there is no guarantee that the final samples satisfy the constraints since the reverse process will go to the next diffusion noisy level from the early stop if the predicted noise is higher than the threshold while the loss is not 0 yet, which is also reflected in the human motion experiment.
2. I get the idea that early stop is determined by the norm of predicted noise from the section 3.2: if the norm is large, then $\mathbf{x_t}$ does not reside in its supposed manifold. But is the reverse direction valid: if the norm is small than some threshold, than $\mathbf{x}_t$ is on the manifold? Is this necessarily true?
3. To summarize above, I found the explanation from methodology section tends to give intuitions about why this algorithm would work instead of guarantees or theoretical analysis, such as a bound on the distance between the $\mathbf{x}_t$ before and after applying loss gradients multiple times, or an upper limit of the probability of samples being out of constraints, etc..

**Questions:**

- In the algorithm, the early termination happens when predicted noise level is higher than the threshold. Shouldn't this be checked at the end of previous step and not applying the gradient from there since the current $\mathbf{x}_{t-1}^*$ already falls off the manifold? Kindly correct me if I am wrong.
- I am not sure if image generation tasks are suitable for this paper. I do not see the constraints but simply comparisons with other baselines for checking the performance of the algorithm.
-  Maybe an ablation study is required here to see how the performance is affected by $\epsilon_{\max}$ instead of choosing $\epsilon_{\max}$ from the samples generated by base diffusion models since it determines when early stop should be.
- Could be more typos but not limited to: from Line 172, ``However, we need to make an to handle inequality constraints``, which affects the reading.

**Limitations:**

Based on the checklist guidelines, the authors claimed the positive and negative societal impact are discussed, but I only found the ``justification`` below the question in checklist is the only place they talked about the impact and I assume this should be mentioned somewhere in the main text or appendix. Limitation is also claimed being discussed in the section 6, and I assume that's mentioned in the second paragraph.

---

> ### Author Rebuttal · Authors · 2024-08-07
>
> We thank the reviewer for acknowledging the significance of our work, noting that our paper is easy to follow, and raising valuable questions for discussion. Here are our responses to your questions:
> 1. Re: guarantee of constraint satisfaction: the reviewer is correct that our method cannot guarantee precise constraint satisfaction. Our method, as well as current SoTA methods such as DPS and DSG, does not claim such a guarantee, because of the general intractability of $p(y|x_t)$ for intermediate Diffusion states, and general non-linear, non-convex constraints this line of work is dealing with. Even without guarantees, the capability of adding constraints provides a training-free technique that allow us to *control* the generation of Diffusion models in many different ways, all using one unified algorithm. In such cases we believe the value of solid empirical experiments cannot be understated, which is why we went the extra mile in implementing multiple quantitative (and qualitative) benchmarks in two drastically different domains and perform careful evaluation in all these tasks, which is unseen in previous SoTA works.
> 2. Re: should the predicted norm threshold be bi-lateral rather than unilateral? While everything is probabilistic here meaning we cannot guarantee $x’$ to be on manifold even if its norm is small, an $n$-dimensional vector with small norm is much more likely a sample from a zero-mean Gaussian than one with large norm. That’s why our threshold is unilateral and only filters large norm epsilons.
> 3. Re: checking threshold before iteration: It’s possible that the added noise $\sigma_t \epsilon_t$ from the previous step results in the predicted noise level within $x_{t-1}$ to be greater than the threshold, thus falling off the manifold. However, in most cases, the falling off of the manifold is not severe, and the current step’s diffusion step will correct for any deviation from the manifold. This may happen over several timesteps $t$ if necessary, and no loss-guided gradient steps will be taken during these timesteps.
> 4. Re: image generation tasks: We will add clarification on this. Such tasks are standard in existing SoTA papers, such as DPS, DSG, LGD-MC, which we borrowed from them. We will make it clear in the main text or Appendix what the constraints exactly are for each task.
> 5. Re: ablation: We’ve added a new ablation study in our one page PDF (Table 4) to illustrate why we choose $\epsilon_\mathrm{max}$ from samples generated by the base diffusion model. The table illustrates that there is a range of effectiveness for $\epsilon_\mathrm{max}$ (roughly between 440-442 for the FFHQ super resolution task), but beyond this range the results do not change too much. Therefore, it suffices to find an $\epsilon_\mathrm{max}$ value within the acceptable range, and a general way of finding this acceptable range is by sampling the base diffusion model.

---

> > ### Comment · Reviewer_rJ33 · 2024-08-11
> >
> > After going through the rebuttals, I decide to increase the score from 4 to 5. Thank you!

---

### Official Review · Reviewer_wQPN · 2024-07-15

**Soundness:** 3
**Presentation:** 3
**Contribution:** 2
**Rating:** 5
**Confidence:** 4

**Summary:**

This paper tackles the task of sampling from diffusion models with additional inference-time constraints. In this setting, synthesis needs to simultaneously follow the diffusion model-defined generative prior as well as a constraint objective. The paper proposes two techniques to achieve this in a robust fashion. First, trust schedules define the number of optimization steps with respect to the constraint that are carried out between each diffusion model generative denoising step. Moreover, state manifold boundaries prevent the model from falling off the diffusion model-defined data manifold during optimization. These techniques enable more robust and stable constraint-guided sampling of diffusion models. The proposed methods are validated on constrained image modeling tasks (superresolution, deblurring, inpainting) as well as several human motion synthesis benchmarks, and favourable results compared to baselines are achieved.

**Strengths:**

**Clarity:** The paper is overall written well, clear to read, and easy to follow.

**Quality:** Overall, the paper is of good quality. The paper is clear, has a detailed discussion of related work, as well as mostly appropriate experiments and ablation studies. I do not see any major technical flaws.

**Originality:** The specific proposed techniques are new and original, to the best of my knowledge. They do represent simple heuristics, though, and I have some concerns discussed below.

**Significance:** I think the paper tackles an important task, generation with inference-time constraints, and it achieves strong performance compared to the baselines. This makes the work generally significant.

**Weaknesses:**

I think there are two main weaknesses:

First, both the trust schedules and also the approach to estimate state manifold boundaries are merely well-motivated heuristics and they both require hyperparameter tuning (how many steps in the trust schedule, $\epsilon_{max}$). For the trust schedules, there is an elaborate derivation that relates approximation errors to state variances (Eq. (9)). But in the end, none of this is tractable and it just motivates the use of a step-dependent optimization schedule, which needs to be chosen fully heuristically. And this schedule also depends on the optimization step size, which needs to be chosen manually, too. The state manifold boundary constraint follows a similar $\epsilon_{max}$ heuristic. In the end, the paper proposes some useful and well-motivated optimization heuristics, but not an overly novel and rigorous framework for constrained diffusion model sampling. This makes the paper somewhat less original.

Second, as the authors pointed out in their related work section, there are many previous papers in the area. However, the authors only compare to DPS as well as DPS+DSG, although there should be further applicable baselines. Most importantly, the authors argue that LGD-MC has no code available. However, I believe reimplementing the LGD-MC method should be quite easy, and a comparison to this work would be very appropriate here. Moreover, maybe LGD-MC could even be combined with, or enhanced by the techniques proposed here. This possibility is not discussed.

**Conclusion:** In conclusion, even though I see some weaknesses and evaluation could be expanded, the proposed heuristics seem useful and the paper is of overall satisfactory quality. Hence, despite my concerns, I am carefully leaning towards suggesting acceptance.

**Questions:**

I have one question: In line 173, the authors write that they replace max with ReLU to have a smooth gradient. What is meant by this? ReLUs do not have a smooth gradient, but it is a step function.

I do not have any additional questions, but here are a few typos:

- line 30: constrained -> constraints
- line 44: it should be "...state manifold of the diffusion model..."
- line 76: I believe the score function misses a $\log$.
- line 182: I believe it should be "LGD-MC".
- line 251: "first first"

One additional comment: In line 65, DDPM and DDIM sampling are described as forms of gradient-based MCMC sampling. I do not think this is generally quite correct. For instance, if one choses a deterministic DDIM setup without noise injection, then this is a regular probability flow without stochasticity, but not MCMC. MCMC typically always has stochasticity with some form of noise injection. Broadly speaking, one can see DDPM and DDIM always as a flow plus a variable MCMC component, but not always as pure MCMC. I would suggest the authors to rephrase this statement.

**Limitations:**

The paper does not have a detailed discussion of its limitations or its societal impact. This did not significantly influence my paper rating, but I would strongly suggest the authors to add a more detailed discussion on both of these aspects to the paper.

---

> ### Author Rebuttal · Authors · 2024-08-07
>
> We thank the reviewer for acknowledging the clarity, technical quality, novelty, and significance of our work, and raising valuable questions for discussion. Here are our responses to the questions:
> 1. Re: heuristics in the method: our paper is heavily inspired by previous works DPS, DPS+DSG and LGD-MC. The focus of these methods, as well as ours, is to design a simple practical algorithm that can be seamlessly integrated into existing Diffusion inference schemes, such that they are *domain-agnostic* as an unified solution for many different domains that utilize the power of Diffusion. With this goal in mind, all these methods strive to distill the insights from theoretical analysis into simple implementations that are computationally tractable and fit nicely into the Diffusion framework. We believe that our results that surpass the current state-of-the-art highlight the practical value of our method.
> 2. Re: comparison with LGD-MC: During the rebuttal period, we implemented LGD-MC with results summarized in Tables 1-3 of our one page PDF, where we find LGD-MC performing on par with DPS and DPS+DSG, outperforming other reported baselines on some tasks while falling behind on others. Trust sampling surpasses LGD-MC in almost all metrics across all tasks, highlighting the strength of our method. Please note that while we try to clearly state all parameters required for our method in the writing of our paper, other algorithms may have additional heuristic parameters that are only clear when the implementations are made publicly available.
> 3. Re: ReLU: thanks for pointing this out. We apologize for this careless error in writing - $\max(0, x)$ just means ReLU here. We will remove the “In practice…” sentence at L173.
> 4. Re: clarity: We will do a careful pass and also explain DDPM, DDIM, and the limitations of our algorithm better.
> Please let us know if this addresses your questions.

---

> > ### Comment · Reviewer_wQPN · 2024-08-10
> > **Reply to Rebuttal**
> >
> > I would like to acknowledge that I have read the authors' rebuttal. Overall, my impression of the paper remains and I maintain my current, positive-leaning rating.

---

### Author Rebuttal · Authors · 2024-08-07

We thank all reviewers for their constructive feedback. We are encouraged by the reviewers’ recognition that our proposed method addresses the important task of Diffusion generation with inference-time constraints, which is widely applicable across many domains. Additionally, our experimental validation spans a diverse range of tasks not seen in previous papers (2D images and 3D human motion), and demonstrates robust performance improvements over recent baselines such as DPS and DPS+DSG. In terms of methodology, reviewers generally agree that our proposed algorithm, which frames guidance as multi-step gradient-based optimization, is both novel and theoretically intuitive.

Reviewers also made a few great suggestions on additional experiments to strengthen the evaluation of this work. We did our best given the tight rebuttal timeframe to implement the following new experiments:
1. We added LGD-MC $n=10$ and $n=100$ baseline results to Table 1 (FFHQ), Table 2 (ImageNet), and Table 3 (motion task table). Trust sampling outperforms LGD-MC on almost all tasks.
2. We performed a new ablation on $\epsilon_\mathrm{max}$, illustrating the robustness of our parameter space.

See our one page PDF for these updated tables. We provide detailed answers to specific questions in the individual responses.

---

### Decision · Program_Chairs · 2024-09-25

**Decision:**

Accept (poster)

**Comment:**

All reviewers are positive  on the paper after rebuttal.
They find the paper well motivated and the method original albeit simple.
The rebuttal addressed most concerns